# Multispectral Remote Sensing Data Application in Modelling Non-Extensive Tsallis Thermodynamics for Mountain Forests in Northern Mongolia

**DOI:** 10.3390/e25121653

**Published:** 2023-12-13

**Authors:** Robert Sandlersky, Nataliya Petrzhik, Tushigma Jargalsaikhan, Ivan Shironiya

**Affiliations:** 1V.N. Sukachev Laboratory of Biogeocenology, A.N. Severtsov Institute of Ecology and Evolution of the Russian Academy of Sciences, Leninsky Prospect 33, 119071 Moscow, Russia; petrzhik.nat@mail.ru (N.P.); ivan.shironya@gmail.com (I.S.); 2International Laboratory of Landscape Ecology, National Research University Higher School of Economics, Pokrovskiy Bulvar 11, 109028 Moscow, Russia; 3Laboratory of Forest Phytocoenology, Botanic Garden and Research Institute Mongolian Academy of Sciences, Peace Avenue 54a, Bayanzurkh District, Ulaanbaatar 13330, Mongolia; tushigmaaj@mas.ac.mn

**Keywords:** ecosystem, exergy, q-index, Tsallis non-extensive thermodynamics, order and control parameters, Landsat 8

## Abstract

The imminent threat of Mongolian montane forests facing extinction due to climate change emphasizes the pressing need to study these ecosystems for sustainable development. Leveraging multispectral remote sensing data from Landsat 8 OLI TIRS (2013–2021), we apply Tsallis non-extensive thermodynamics to assess spatiotemporal fluctuations in the absorbed solar energy budget (exergy, bound energy, internal energy increment) and organizational parameters (entropy, information increment, q-index) within the mountain taiga–meadow landscape. Using the principal component method, we discern three functional subsystems: evapotranspiration, heat dissipation, and a structural-informational component linked to bioproductivity. The interplay among these subsystems delineates distinct landscape cover states. By categorizing ecosystems (pixels) based on these processes, discrete states and transitional areas (boundaries and potential disturbances) emerge. Examining the temporal dynamics of ecosystems (pixels) within this three-dimensional coordinate space facilitates predictions of future landscape states. Our findings indicate that northern Mongolian montane forests utilize a smaller proportion of received energy for productivity compared to alpine meadows, which results in their heightened vulnerability to climate change. This approach deepens our understanding of ecosystem functioning and landscape dynamics, serving as a basis for evaluating their resilience amid ongoing climate challenges.

## 1. Introduction

The exploration of living systems through the lens of thermodynamics and statistical mechanics traces its roots back to Alfred J. Lotka’s pioneering works [1,2]. Over time, the study of the environment, including biota, has bifurcated into two primary trajectories: climatic and ecosystem (except at the level of individual organisms). Thus, we owe the modern understanding of energy transformation at various levels to physicists, meteorologists, and climatologists and biologists, ecologists, and mathematicians. The climatic approach, originating in the simple entropy model of climate system performance in the late 1970s [3,4,5,6], contrasts with the ecosystemic perspective, based on biophysical studies by Harold Morowitz [7,8], ecological modelers, like Yuri Svirezhev [9,10,11] and Sven Jørgensen [12,13], and ecological theorists, like Robert Ulanowicz [14,15]. Consequently, the energy balance of ecosystems within the thermodynamic framework was explored through these distinct paradigms, namely climatic modelling and ecological modelling.

Ecologists have prioritized the development of community structure and organization in energy conversion processes. Concurrently, the concept of energy quality emerged, emphasizing its ability to perform useful work and sustain living matter’s structure. In the late 1980s, terms, such as emergy [16,17] and subsequently exergy [18,19,20], found their way into environmental modelling. While emergy faced limited recognition due to its economic orientation [21], exergy analysis firmly established its position in studying various organizational levels of living matter, from local communities to the entire biosphere. The fundamental principles of this approach are most comprehensively described in the works of Sven Jørgensen and his co-authors [22,23,24]. A thorough examination of the practical applications of the exergy method in the modelling of energy within ecosystems is available in the comprehensive review authored by Nielsen and colleagues [24]. The thermodynamic basics of the functioning of living matter were elucidated by Axel Kleidon and Ralph Lopez in their seminal paper “Non-equilibrium thermodynamics and the production of entropy: Life, Earth, and beyond” [25]. Subsequently, Axel Kleidon made substantial contributions to the understanding of the role of vegetation in climate systems, particularly from the point of view of entropy [26,27,28]. With advances in the measurement of biophysical parameters of vegetation cover through certain methods, like eddy covariance, projects dedicated to assessing entropy production and self-organization in specific ecosystems have emerged [29,30]. In summary, the current state-of-the-art of ecosystem thermodynamics is marked by the predominance of meteorological and ecosystem-focused approaches. Notably, a review of key publications from both perspectives [23] vs. [25] indicates that researchers tend to embrace one approach while largely neglecting the other, possibly due to differences in research objectives. Nonetheless, the division between the strategies employed by physicists and ecologists in assessing the thermodynamics of ecosystems appears noticeable. We believe that this division will inevitably be bridged as a result of the accumulation of valid eddy covariance field data, the expanding repository of remote sensing data, and the continued development of modelling methodologies.

With the expansion of the biophysical measurement network, particularly eddy covariance, through initiatives, like Fluxnet [31], researchers now face challenges demanding a nuanced understanding of ecology and adeptness in modelling community structures. Consequently, a third approach has gained traction, emphasizing the evaluation of landscape effects as a complex of ecosystems in terms of energy fluxes [32,33,34,35]. Such studies are already being conducted in line with the influence of the structure of landscape cover as a set of ecosystems on energy flows, meaning that they meet the tasks of landscape ecology. In turn, in landscape ecology in recent years, there has also been an obvious increase in interest in describing the spatial structure of the landscape through thermodynamics, for example, in the works of Samuel Cushman [36,37], Peichao Gao, and Hong Zhang et al. [38,39,40,41]. However, the widespread installation of eddy covariance measurement stations is hindered by their substantial temporal and financial investments, limiting their applicability to specific areas and local conditions. Thus, similar data should be judiciously applied, tailored either to specific ecosystem functioning or regional generalizations, integrating various scales of additional data into models. Remote sensing assumes a pivotal role, offering unique insights into vegetation cover properties, including albedo, vegetation indices, and gross primary production. A highly detailed review of the use of remote sensing in estimating energy balance components can be found in the work of Masudur Rahman and Wanchang Zhang [42]. Nevertheless, assessing entropy flows and the useful work of vegetation cover, especially in the thermal (IR) range, encounters challenges due to the incomplete radiation spectrum, infrequent high-quality satellite images caused by unfavorable weather conditions, and the lack of a methodological foundation enabling direct multispectral measurements for an ecosystem’s work parameters calculations.

When using multi- and hyperspectral information, the radiation emitted by objects serves as a repository of data, with the energy spectrum corresponding to satellite spectral bands [43]. Information about the physical and biochemical processes occurring within ecosystems, upon which all indices are based, is encoded in the reflected energy ratios across different spectral bands [44]. By comparing incoming and outgoing radiation values precisely at the moment of capture, the absorbed solar radiation in each band can be estimated. This systematic structure, akin to a “solar receiver”, can be described using information measures. Specifically, emitted solar radiation, entropy flux, and the Kullback information increment (Kullback–Leibler divergence/relative entropy) can be calculated through the reflection–absorption ratio in various spectral bands. These measure characterize the non-equilibrium between the reflected and incoming spectra of solar energy (solar constant). Consequently, based on these measures, the components of absorbed solar energy can be evaluated, including Gibbs free energy (exergy), bound energy, and the increment of internal energy. This approach was founded by Yury Svirezhev in the early 2000s [45,46], with subsequent practical implementation by Yury Puzachenko and colleagues using various multispectral [47,48] and hyperspectral [49] remote sensing systems. However, it is important to note that the estimates derived from this approach are relative, despite being measured in standard energy units (W/m²) from multispectral reflectance. Multispectral reflective variables do not cover the entire incoming solar radiation spectrum as they are measured in relatively narrow spectral bands. Consequently, the results obtained using this approach cannot be directly compared with eddy covariance measurements from field studies.

The advancement of this approach was linked to expanding the standards of the Boltzmann–Gibbs–Shannon (BGS) entropy model to non-equilibrium statistical mechanics (NSM) [50,51]. Constantino Tsallis [50] admitted that the strong interaction in thermodynamically abnormal systems changes the picture so much that it leads to completely new, independent degrees of freedom, and to a completely different statistical physics of the non-Boltzmann type. Thus, NSM, in contrast to BGS, incorporates nonlinear interactions between system components and is particularly suited for living systems due to their non-equilibrium state and complex nonlinear interactions. Tsallis [50] introduced the q-index (power coefficient of deformation) into the entropy equation. The q-index characterizes the scale of correlations between the system’s components and usually resides between −1 and 2, where 1 is an equivalent of Boltzmann–Gibbs entropy. Additivity is described by q as follows: for q > 0, the q-entropy, like the BGS entropy, is greater, resulting in a smaller difference in the logarithms of the probabilities of neighboring energy levels (neighboring classes) in the rank distribution, while for q < 0, the opposite is true. In other words, a change in the parameter q also reflects the presence of stable positive correlations between elements in the system. The power-law form of interaction between elements leads to the fact that there is a region of states of the variable y in which it is practically insensitive to changes in x and vice versa. As a result, any element can simultaneously interact with many others without actually distinguishing between their states [51]. If we add a time coordinate to space, then power relations create the basis for “memory”, since over a certain time interval the magnitudes of interactions will change slightly. Accordingly, a q-index exceeding one indicates greater internal correlations and higher organization within the system. Conversely, a q-index below one suggests a non-organized system. One of the fundamental characteristics of Tsallis entropy is its non-additivity (unlike Renyi entropy, for example), implying that the system’s entropy does not equal the sum of its subsystems’ entropies.

We obtain the parameter q based on a comparison of the maximum possible entropy at each point (i.e., complete chaos of the system) and the lowest entropy achieved at one of the values of q—i.e., with the greatest sensitivity of the index to the amount of order in the system. Thus, in this case, taking into account the general interpretation of q as a measure of additivity in the system, it can be assumed that the value of q giving the lowest entropy reflects the degree of interdependence of elements within a particular display of landscape cover through reflected solar radiation. Thus, the assessment of landscape cover functioning parameters based on NSM emphasizes the non-equilibrium regions of space. System order parameters (invariants) lag in adapting to the influence of control parameters. Examples of such regions include boundaries between different ecosystems, areas undergoing active conversion due to exogenous factors, and artificial ecosystems. Our preliminary experiments on real ecosystems [52,53] demonstrate the validity of computations performed with thermodynamic variables of landscape cover calculated from multispectral data within the framework of NSM.

This paper illustrates the assessment of thermodynamic parameters via NSM in mountain forest ecosystems in Northern Mongolia. Forest landscapes occupy approximately 7% of the entire Mongolian area. About 80% of this area is covered by mountain forests that are located in the north of the country. Under the extreme/sharp continental climate conditions there, these landscapes are often characterized by permafrost. Therefore, they play a key role as a factor maintaining permafrost cover for the surrounding area, respectively, a significant amount of carbon is conserved by mountain forests working together with soils. Mountain forest ecosystems importance rises because of the accelerated rate of global warming/changing and with regional area aridification simultaneously. Specifically, such ecosystems perform a buffer function restraining desertification and, according to some estimates, can accumulate about 50% of Mongolia’s greenhouse gas emissions [54]. Mongolian mountain forests are experiencing intense/severe pressure caused by grazing, logging, and fires [55]. Mountain forests, characterized by diverse relief, varying steepness, slope orientation, and abundant vegetation species composition, pose significant research challenges. Despite methodological and logistical hurdles, the integration of remote sensing data, digital elevation models, and field data offers unprecedented insights into these ecosystems’ energy dynamics. The Tsallis model, applied in this context, delves into the intricacies of NSM and self-organization, enhancing our understanding of biosphere operations at the landscape cover level. This integration not only enables the calculation of energy metrics for mountain forests but also transforms our comprehension of their structure and organization. The strongly continental climate of the study area contributes to the regular accumulation of remotely sensed information, which, in combination with expeditionary research (the Joint Russian–Mongolian Complex Biological Expedition of the Russian Academy of Sciences and the Mongolian Academy of Sciences), creates a unique opportunity to expand our understanding of land cover as a complex self-organizing system.

## 2. Materials and Methods

### 2.1. Study Area

The study area is situated within the Horidol-Saridag Strictly Protected Area (50.90 N, 99.88 E), along the ridge that separates Lake Khövsgöl and the Darkhad Valley (Figure 1). This region falls under the Ulaan Taiga Specially Protected Areas designation, including three additional areas: Ulaan Taiga Strictly Protected Area and Tengis–Shishged National Park [56]. These regions are located in the northernmost aimag (province) of Khövsgöl in Mongolia. Elevations within this study area range from 1600 m in the Darkhad depression to 3000 m above sea level at the Khuern Uul mountain. The predominant geological composition of this region consists of limestone carbonates from the Cambrian period. Geologically, the area underwent at least two glaciations during the late Pleistocene [57].

The study area experiences an extremely continental climate characterized by brief, humid summers and prolonged, dry winters. Mongolia’s landlocked southern position, numerous internal and adjacent mountain ranges, and the Siberian anticyclone influence these climate patterns. The Siberian high amplifies atmospheric pressure, leading to calm, sunny weather, lower temperatures, and reduced precipitation. The combination of low atmospheric humidity in spring, winter freezing of moisture in upper soil horizons, and windy conditions slows down soil warming during the anticyclone dissolution period [58]. Based on data from the nearest weather station located in Renchinlhumbe somon (1573 m above sea level), situated 30 km from the research site and spanning from 1975 to 2015, the average annual temperature is recorded at 6.9 °C. Annual precipitation averages 263 mm (±0.1 SE) and exhibits considerable variation, occasionally reaching up to 400 mm/yr. The summer period from July to August witnesses the highest precipitation, accounting for up to 80% of the total. Vegetation cover [59,60] on the lower and middle slopes primarily comprises montane taiga forests, dominated by larch (*Larix sibirica*) or pine–larch (*L. sibirica, Pinus sibirica*) communities. These forests grow over organic soil exhibiting permafrost characteristics. As elevation increases, alpine meadows dominated by *Carex amgunensis* and *Festuca altaica* emerge on mineral soils, eventually transitioning to barren rocky peaks at higher altitudes.

### 2.2. Calculation of Thermodynamic Variables

The conversion of solar energy within ecosystems was comprehensively examined by analyzing various thermodynamic parameters. Data for these parameters were extracted from the multispectral Landsat 8 OLI (Operational Land Imager) TIRS (Thermal Infrared Sensor), manufactured by NASA Goddard Space Flight Center (Gilbert, US) collections, providing a spatial resolution of 30 m. Twenty-three cloudless scenes captured between 2013 and 2021 were meticulously scrutinized. Detailed information, including average values of incoming solar radiation, albedo, and surface temperature for the research area, is presented in Table 1.

The initial values of reflection brightness registered by the multispectral scanner were meticulously transformed into reflected solar radiation within the landscape. These conversions adhered to standard calibration coefficients unique to each measurement system and band. Initially, solar radiation reflected by Earth and detected by the Landsat satellite in each band was calculated using calibration coefficients [61]. Subsequently, the incoming radiation to Earth’s surface in each band was estimated. This involved correcting the extraterrestrial solar irradiation (ESUN) for each band, considering the “Earth–Sun” distance and the Sun’s position during imaging. The assessment of reflected and emitted radiation by Earth (in W/m^2^) for each spectral range was then obtained. These values were multiplied by the average value or the corresponding range (in μm) to determine incoming and Earth-reflected solar radiation. The overall energy flux equaled the sum of flux in each spectral range. The energy absorbed by the Earth’s surface was computed as the difference between incoming and reflected energy. The conversion of the thermal infrared band into energy units indicated the heat flux from the Earth’s surface, which was further calculated into surface temperature values using calibration coefficients.

The energy balance of components was calculated utilizing the method proposed by Yury Puzachenko and co-authors [47,48]. Following the principles of standard Boltzmann–Gibbs’ thermodynamics, the system’s energy balance, absorbing energy, consisted of free energy (potential for performing useful work in the system, or exergy), bound energy (energy incapable of doing work), and internal system energy increments, as follows:B = Ex + STW + U (1)
where B represents the balance of absorbed energy, Ex is exergy, SW represents bound energy, and U signifies internal energy. Exergy is the energy that can be converted into useful work. Regarding ecosystems, exergy closely relates to the maintenance of the water cycle. According to the founders of the idea, Jorgensen and Svirezhev [22], exergy also includes energy expenditure for photosynthesis; however, in our studies based on large field material, the exergy that is calculated using Landsat multispectral channels does not correlate with biological productivity, but rather the opposite (we have shown that exergy is almost entirely at its maximum for climax forests with minimal production). Bound energy is the energy that is dispersed into the environment together with heat flux and entropy. In fact, this is a flow of “waste” energy incapable of performing work useful for the system. Internal energy increment means energy accumulation by organics, like peat, and carbon itself in the soil. This energy is spent on maintaining chemical bonds, interspecies interaction, and “friction” between the elements of the system [22]. The proposed approach to converting multispectral measurements differs from the approach that relies on the use of various classical semi-empirical vegetation indices (for example, the Normalized Difference Vegetation Index, NDVI, which we also use here as a generally accepted tool for assessing vegetation productivity, since it is closely related to net primary productivity, NPP [62], which we, of course, also consider in this work).

In open non-equilibrium systems, Gibbs free energy is equivalent to exergy, which is the energy that can be extracted from the system while maintaining a balance with the environment. The non-equilibrium of solar energy conversion in the system was estimated by the deviation of the reflected solar energy spectrum from the hypothetical equilibrium spectrum. This was assessed using the Kullback information increment (Kullback–Leibler divergence, Kullback information). The Kullback information was evaluated for multispectral imaging as follows:
(2)
$$K=\sum_{n=1}^{n}p_{v}^{out}ln\frac{p_{v}^{out}}{p_{v}^{in}}$$

where 
$p_{v}^{in}=e_{v}^{in}/E^{in}$
 is the ratio of incoming energy (
$e_{v}^{in}$
) in the spectral band (*v*) and the total incoming energy (solar constant) in all bands is (*E^in^*), and 
$p_{v}^{out}=e_{v}^{out}/E^{out}$
 is the ratio of outgoing (at-sensor reflected radiation) energy (
$e_{v}^{out}$
) in the spectral band (*v*) and the total incoming energy (*E^out^*).

The Kullback information is zero if the outgoing radiation spectrum is similar to the incoming one. If it is greater than zero, it indicates that information has increased in the receiver (in this case, the vegetation cover), and the reflection surface is non-balanced concerning the solar radiation spectrum. Thus, the absorbed solar energy possesses the potential to perform useful work, i.e., exergy. Exergy (Ex), calculated through the increment of information, is determined as follows:
(3)
$$\mathrm{Ex}=E^{out}(K+\ln A)+B$$

where *A = E^in^/E^out^* is the albedo, *B = E^in^ − E^out^* represents ecosystem energy balance (energy absorbed by the ecosystem). Useful work in an ecosystem is commonly interpreted as the energy cost to maintain the local water cycle (evapotranspiration) and ensure biological output production. Our studies analyzing spatial–temporal variations in exergy have indicated that exergy measured on multispectral data is loosely linked to ecosystem productivity and primarily governs the moisture pathway from soil to the atmosphere [47,48]. Meanwhile, Kullback information is closely associated with biological productivity and can be evaluated through Landsat vegetation indexes, such as NDVI.

Another parameter describing the non-equilibrium system is the reflected solar irradiance entropy (*S^out^*). This measure demonstrates the variety in the reflected radiation spectrum and indicates the system’s organization in energy conversion. Entropy is estimated as follows:
(4)
$$S^{out}=-\sum_{n=1}^{n}p_{v}^{out}lnp_{v}^{out}$$


Bound energy (STW) represents the dissipated environmental energy incapable of useful work and is determined by the product of the reflected solar irradiance entropy (*S^out^*) and heat flux (TW), as follows:
STW = *S^out^* × TW.(5)

The final step in achieving balance is computing the internal system energy growth (U). This energy is typically associated with processes accumulating energy in the system, predominantly in the form of organic matter (e.g., peat accumulation, humus formation), as follows:
DU = B − Ex − STW
(6)

The thermodynamic parameters above profile solar energy conversion by the ecosystem within the framework of classical Boltzmann statistical mechanics for non-equilibrium systems. In these systems, components have linear interactions at the microscopic level, as follows:dy/dx = a, y = a + x(7)
and, respectively, the inverse is true, i.e., x = a − y
where x and y are abstract components of the system. So, for example, we can consider the spectral bands of reflected solar radiation as a reflection of the components of the system. Then, if the photosynthetic system carried out linear interactions, then one could count on an additive effect when one of the components is changed. In nonlinear components (which include landscapes), changes can have a non-additive effect.

However, in systems far from equilibrium, including complex and non-stationary ones, components may engage nonlinearly, particularly exponentially. In such cases, the transition of the system state at the macroscopic level can be described as follows:
(8)
$$dy/dx=y,\;y=e^{x}\;and\;relatively,\;the\;reverse\;is\;true\;x=\ln y,$$
Equations (7) and (8) show that y(x) = 1 if x = 0.

Both functions can be generalized through the power function proposed by Constantino Tsallis [50]:
(9)
$$dy/dx=y^{q},\;y=[1+(1-q)x{]}^{\frac{1}{\left(1-q\right)}}\equiv e_{q}^{x},\;e_{q=1}^{x}=e^{x}$$

and the opposite:
(10)
$$y=\frac{x^{1-q}-1}{1-q}\equiv \ln_{q}x,\;\ln_{1}x=\ln x\;,where\;x>0$$


In a similar system, entropy is referred to as Tsallis entropy and defined as follows:
(11)
$$S_{q}=\frac{1-\sum_{i=1}^{n}p_{i}^{q}}{q-1}$$

where *p_i_* represents a discrete set of probabilities *i*, 
$S_{q}^{max}={ln}_{q}n$
 is the entropy maximum at *p_i_* = 1/*n*, and *n* is the class number. *q* is the deformation parameter (*q* = 1 − 1/*r*, where *r* > 0 is a correlation coefficient). A higher *q* signifies greater internal correlations and a more organized system. The q-parameter provides information about the different contribution ratios of the components in the energy balance of the ecosystem. Deviation from stationarity indicates incomplete transformations related to species interactions and irregular effects of external factors (control parameters). Hence, the q-parameter can be considered a measure of stationarity. When *q* deviates from unity, the system either exhibits signs of degradation (at *q* < 1 *q* < 1) or demonstrates non-stationary processes typical of the active development of “living systems” [63]. Accordingly, landscape cover functioning parameters based on non-extensive Tsallis thermodynamics should encourage the assessment of the heterogeneity factor space. Moreover, they should identify active zones highlighting where specific processes differ from the main stationery in the work of invariants. Examples of such zones include boundaries between different ecosystems, territories undergoing active changes associated with exogenous factors, and artificial ecosystems.

The Kullback information increment assesses the distance between a stable and non-stable system in the Tsallis system, as follows [64]:
(12)
$$K_{q}=\sum_{i=1}^{k}p_{i}\frac{[p_{i}^{out}/p_{i}^{in}{]}^{q-1}-1}{q-1}$$


The exergy of solar radiation and another balance components in the Tsallis system (Ex_q_) is calculated based on the Kullback information increment (*K_q_*) as well as the classical thermodynamics model (BGC).

### 2.3. Order Parameters and Control Parameters of the Thermodynamic System

The spatial–temporal variation in these thermodynamic parameters was analyzed using the concepts of invariants and order parameters. This approach enables the description of the system state through a limited number of factors.

The concept of order parameters, as proposed by Hermann Haken in 1983 [65], gained prominence in the scientific literature through the work “Principles of Brain Functioning” [66]. Here, the identification of order parameters via principal component analysis (PCA) was established. Yury Puzachenko later equated H. Haken’s “order parameter” with Victor Sochava’s (1978) “landscape invariant” [67]. Identification of these invariants was based on multispectral remote sensing data, which were further employed to identify stable spatial structures of thermodynamic or reflectance systems [68,69]. Spatial–temporal fluctuations in variables were generalized using PCA and the varimax normalization method for each variable (23 terms). They were then classified into seasonal (winter, spring, etc.) invariants of the thermodynamic system.

In theoretical terms, changes in order parameters are determined by control parameters in thermodynamic systems. These control parameters, often termed external factors, include climate, weather conditions, relief, and the state of vegetation cover. In this study, a set of vegetation properties acquired through fieldwork, along with morphometric characteristics of the terrain at different hierarchical organization levels, were considered as control parameters. According to the hierarchy paradigm, ecosystems can be perceived as a complex of subsystems at varying scales with distinct properties. These hierarchical levels of relief organization were differentiated through spectral analysis of the open-access digital surface model SRTM [70] with a spatial resolution of 30 × 30 m (see Figure 2). This approach was proposed by Donald Turcotte [71]. Linear dimensions were used to best characterize the identified relief at 150, 570, 1110, 1380, 1890, and 3000 m. Standard morphometric characteristics reflecting the relief’s influence on heat and moisture management, such as steepness, aspect, minimum and maximum curvatures, and concave (negative) and convex (positive) curvatures, as well as plan (horizontal), profile (vertical), and general (total) curvatures, were evaluated within each hierarchical level.

The impact of terrain on energy conversion was assessed by employing multiple regression analysis on the invariants obtained from morphometric characteristics. Field measurements of ecosystem properties were conducted using the line transect method. A 1-kilometer-long transect with control sites every 20 m was established along the steep southern slope, ranging from 2000 to 2380 m asl (Figure 2). Comprehensive descriptions of the soil and vegetation were recorded for almost all sites. Field characteristics, including stand basal area and canopy percentage projection to calculate the leaf area index (LAI), were estimated. Furthermore, fresh phytomass was sampled from a 25 × 25 cm surface square at each site in triplicate, and the dry phytomass weight was determined. The transect spanned pine-dominated forest communities at the base and middle parts of the slope (2000–2200 m asl) and extended to alpine meadows (2200–2350 m asl) before transitioning into barren rocky peaks at higher altitudes.

## 3. Results

### 3.1. Order Parameters of Thermodynamic Variables

Upon employing PCA, three distinct order parameters emerged for each thermodynamic variable. The initial parameter delineates the winter “mode” (October–March) of ecosystems in the examined region, the second characterizes the summer “mode” (June–September), and the third embodies the transitional “mode” (April–October). It is noteworthy that the method used did not identify a transition “mode” for the heat field of the ecosystem. Table 2 presents the explained variance ratio (EVR in %) of absorbed solar energy and energy balance components, as described by PCA factors. It is essential to note that the EVR distribution between factors depends on the number of scenes for a specific season. Consequently, the primary variance ratio (65–70%) for all variables, except internal energy increments, is attributed to the cold (winter) period. In contrast, the summer invariant accounts for 13–16%, and the transition invariant represents 5–6%.

Significantly, the ratio of internal energy increments is significantly determined by the winter invariant (54%) and the summer invariant (25%). Together, these three seasonal order parameters collectively elucidate approximately 85–90% of the variations in energy balance components, including NDVI, surface temperature, and the entropy of reflected solar radiation. Furthermore, the EVR for information implementation and q-index, explained by the summer invariant, exceeds that of other variables, standing at 25% and 22%, respectively. This places the EVR for these parameters in close proximity to the EVR for the internal information increment. Additionally, the EVR of informational–thermodynamic parameters described by these three invariants is slightly lower than that for balance components. This observation implies that ecosystems primarily exist in two stable states and one transitional mode when surface temperature approaches zero. The winter “mode” of ecosystems is characterized by minimal vegetation cover performance, acting as the primary converter of solar energy. Henceforth, the active state of vegetation cover will be the focal point of consideration.

Figure 3 illustrates the territory’s differentiation based on the degree of vegetation cover development. Exergy, information increment, and q-index are minimal on open grounds/surfaces. In contrast, the fluctuation of bound energy follows a complex pattern contingent on landscape/relief. The summer order parameters obtained (detailed in Table 2 for 10 variables) underwent PCA to delineate subsystems within the thermodynamic system responsible for various processes.

Throughout the growing season, three order parameters of the thermodynamic system, depicting its state, were identified. Despite the description of seasonal order parameters, some fluctuations in initial variables remained unaccounted for. Therefore, a comprehensive analysis of all initial thermodynamic variables throughout the growing season (10 variables for each of the 9 periods) was conducted. Similar results were obtained: three order parameters explained 78% of variations in thermodynamic variables. The first parameter accounted for 43.6%, the second for 26.7%, and the third for 8.1%, respectively.

These highlighted order parameters of the thermodynamic system are determined by variables from three functional subsystems: “absorbed energy and energy costs to evapotranspiration”, “temperature, bound energy and entropy”, and “structural and productional (the increment of information, q-index, and NDVI)”. These subsystems elucidate three distinct processes governing solar energy conversion by the landscape of the area, as follows:The evapotranspiration process: this determines the absorption of solar energy by vegetation cover and the transfer of moisture from the soil into the atmosphere;Energy dissipation: this occurs in the atmosphere and is reflected in the heat field. This can be detected through heat surface capacity, level of lighting/exposition, and atmospheric stratification, as assumed;The production process: this is linked to the non-stationarity of vegetation cover and identified through non-stationarity and degree of self-organization in vegetation cover.

### 3.2. Relief as a Control Parameter of the Thermodynamic System

Relief emerges as a pivotal factor steering the intricate functioning of landscape cover. Through meticulous regression analysis, we explored the interconnection between summer order parameters for each thermodynamic variable and morphometric landscape features, reflecting heat and moisture distribution across varying scales (hierarchical levels). The outcomes, meticulously compiled in Table 3, spotlight a coefficient of determination ranging from 30% to 40% for albedo, absorbed solar radiation, and its balance components. Notably, surface temperature exhibited the strongest correlation (R^2^—0.63), followed closely by the information increment (R^2^—0.51) and exergy (R^2^—0.48). However, relief’s influence on the spatial variation in reflected solar radiation entropy was nearly negligible (R^2^—0.1).

The substantial impact of relief surfaced predominantly in structures spanning 570–1380 m in linear size, roughly corresponding to the average slope length. The first parameter to heighten as luminosity increased was bound energy, indicating the dissipation of energy into the atmosphere. Figure 4 illustrates the summer order parameter values for solar radiation exergy predicted via morphometric features, along with the corresponding model residuals.

The contribution of morphometric characteristics to the thermodynamic system’s order parameters was likewise scrutinized. The analysis delved into three sets of order parameters, each dissecting distinct facets: the total absorption of solar energy and exergy, bound energy (a product of heat flux and entropy), biological productivity, and structural variables encompassing the increment of information and q-index.

The regression model sheds light on 48% of the explained variance ratio of the first set (Figure 4a). Notably, the cumulative EVR of the components of the first set for relief structures around 580 m in linear size diminished with decreasing height above sea level and intensified with heightened curvature, a characteristic often found in concave slopes within our research area. Additionally, the steepness of slopes measuring more than 1890 m influenced the total absorption of solar energy and exergy. Figure 4b,c depicts the predicted values of total absorption of solar energy and exergy through relief. These values reveal that solar energy absorption and exergy peak in the valleys and plains of the north-western area, whereas they dip significantly on the convex surfaces of ridges. The model exhibited reliability, as evident from the normal distribution of residuals (Figure 4c). Positive residuals on concave and gentle slopes indicate overestimation of absorption and exergy by relief, while negative residuals on steep southern slopes point to underestimation. Residuals hovering around zero (yellow colour) correspond to valleys.

Morphometric characteristics accounted for 54% of the explained variance ratio of the second set (Figure 4d). Sunlight from the south positively impacted the bound energy of relief structures measuring around 580 m and 1110 m in linear size. Unlike the first set, the bound energy of structures measuring 570 m decreased with increased curvature. Consequently, the terrain-based model underrated warming on western slopes and overrated cooling on northern slopes. The model illuminated 33% of the explained variance ratio of the third set using morphometric characteristics (Figure 4g). The parameter values dwindled with ascending altitude and heightened with southward and eastward solar irradiance. These trends were particularly noticeable for relief structures measuring 570 m in linear size. Consequently, relief prognosticated elevated values of biological productivity and structural variables in the valleys and plains, while lower values corresponded to ridge surfaces (Figure 4h). The distribution of residues mirrored the distribution of the third set itself (Figure 4g), with predicted parameter values surpassing actual values on flat plains.

In summary, the intricate topography of the Arctic landscape significantly influences the absorption, distribution, and utilization of energy within these ecosystems. Valleys and plains exhibit distinct energy dynamics compared to ridges, highlighting the nuanced interplay between relief and the delicate balance of energy in these Arctic regions.

### 3.3. Vegetation as a Control Parameter of the Thermodynamic System

The Pearson correlation coefficient (r) was employed to analyze the relationship between summer invariants of thermodynamic variables and the characteristics of vegetation and soil cover, as detailed in Table 4. These characteristics were meticulously measured at various points along the transect, with a focus on the most robust correlations. Primarily, the mass of vegetation, particularly wood mass, amplifies solar energy absorption and elevates evapotranspiration costs. Consequently, this vegetation parameter diminishes bound energy and the increment of internal energy. These deductions are applicable to vegetation mass computed from field data, including basal stand area (BSA), leaf area index (LAI), stand canopy density, and phytomass. Among these variables, only LAI demonstrates a direct association with vegetation productivity (NDVI); higher LAI values correspond to elevated NDVI levels. Simultaneously, an increase in woody vegetation mass augments the information increment while reducing entropy and the q-index. Surface temperature exhibits a negative correlation with the projective cover of the moss–lichen layer, subsequently correlating with stand canopy density and phytomass. Additionally, higher soil pH levels correlate with increased absorption of solar energy, evaporation costs, and NDVI. Consequently, the bound energy, increment of internal energy, entropy, and q-index decrease with rising soil pH levels. The influence of organic layer thickness on thermodynamic parameters is relatively weak (r = 0.3), akin to the impact of BSA and LAI.

It is logical to posit that fluctuations in thermodynamic variables uncontrolled by relief are influenced by vegetation properties. Consequently, the interaction between vegetation and the order parameters of the thermodynamic system, as well as with residuals based on relief, is explored through Pearson correlation coefficients (r) in Table 5.

While the correlation between ecosystem properties and order parameters is comparatively weaker than the correlation of individual variables with ecosystem properties, their interaction pattern remains consistent. Notably, wood vegetation mass enhances solar energy absorption, exergy, and productivity, while diminishing bound energy. Moreover, the correlation between residuals and solar energy absorption values surpasses that between residuals and field data in general. Field data correlations with their respective values are stronger than those between model residuals and other parameters. These intricate relationships between order parameters and key measured vegetation properties, such as BSA and stand canopy density, are graphically represented in Figure 5.

### 3.4. Types of Thermodynamic Systems

The conversion of solar energy is intricately linked to the distribution of order parameters at each point (a pixel in our case) within a system. Consequently, the study area underwent a dichotomous classification based on the ratio of three order parameters, resulting in diverse ecosystems and land cover classes (Figure 6a). This led to the identification of six primary types of thermodynamic systems: larch forest, pine forest, meadow steppe, alpine meadow, colluvial material, and rock formation. Types are presented in accordance with the “Vegetation map of Mongolian People’s Republic” [59] and the “Ecosystems of Mongolia. Atlas” [60] and interpreted based on our field geobotanical descriptions and visual interpretation of medium-resolution satellite images. Subdividing these into more detailed subtypes proved challenging, as it necessitated a deeper understanding of the territory for accurate interpretation, and, thus, was not pursued. The types of thermodynamic systems, determined by the distribution of order parameters, are illustrated in Figure 6b. Rock formations and colluvial materials exhibit minimum values for all parameters. Forests, on the other hand, display maximal values of exergy and absorption, while their dissipation of energy through heat flux and entropy into the atmosphere is minimal (second parameter). Meadow steppe and alpine meadows are characterized by low absorption and exergy values, counterbalanced by maximum energy dissipation. An important observable result is a clean separation of manifolds describing different types of landscape projected on the basis of order parameters’ space.

Figure 7 presents the seasonal patterns of key thermodynamic parameters: exergy, active surface temperature, vegetation productivity index (NDVI), and q-index. The seasonal trends of exergy (Figure 7a) and incoming solar radiation, along with the active surface temperature (Figure 7b), closely mirror each other. Throughout the year, alpine meadows and rocks consistently exhibit the highest temperatures. Forest types demonstrate peak exergy and minimal temperatures year-round. Meadow steppe experiences a surge in exergy during summer, aligning with forest types due to the growing season. Seasonal variations in NDVI (Figure 7c) and q-index indicate an active vegetation period lasting from early June to mid-September, approximately a hundred days. The q-index serves as a vital quantitative indicator, delineating the commencement (q > 1) and conclusion (q < 1) of the active period in the ecosystem. Meadow steppes exhibit peak productivity between June and September. Alpine meadows and pine forests display similar productivity dynamics, whereas larch forests exhibit relatively low annual productivity. Meadow steppes and alpine meadows (Figure 7d) demonstrate the highest levels of self-organization (maximum q-index) from late June to the end of September. Forest types exhibit lower self-organization, with a more prolonged period of non-equilibrium states. Intriguingly, self-organization in larch forests drops below 1 in mid-summer. A preliminary hypothesis attributing this to excess heat supply (Figure 7b) and moisture deficiency is proposed. However, this hypothesis requires refinement and validation through comparison with weather conditions.

Figure 7e shows the seasonal variation in the Kullback information increment. It clearly demonstrates the imbalance between pine forests in winter and larch forests in summer. This is due, firstly, to the fact that larch forests remain bare in winter, while pine forests retain their needles. At the same time, pine forests are confined to the southern slopes, and this is probably due to their high disequilibrium and vulnerable state in the autumn. At the same time, in summer, their level of disequilibrium is not high, which correlates with their low level of productivity (Figure 7c). Meadows and steppes, as expected, have a minimum disequilibrium in winter and a maximum in the snow-free period. Figure 7f shows the seasonal dynamics of the entropy of reflected solar radiation. Maximum entropy is characteristic of surfaces with poorly developed vegetation and decreases as the mass of woody vegetation increases. Thus, we obtain the following picture: pine and larch forests maintain maximum exergy throughout the year with minimal entropy, average levels of organization, disequilibrium, and q-index, and a low level of productivity. Meadows and steppes with maximum entropy have high productivity and disequilibrium during the period of active work. Thus, different strategies for using solar energy are obvious: high exergy with low self-organization and disequilibrium and productivity in forests and low exergy with high disequilibrium and self-organization in meadows, but in a short time. In general, the larger the q-index, the lower the exergy and entropy, but the greater the Kullback information. From the perspective of the theory of self-organization, we find that as the organization grows, the “useful” work of the system (exergy) decreases. Thus, it can be assumed that for the class of systems under consideration, self-organization, described by the growth of the q-index, increases the efficiency of the system in that it reduces energy costs for supporting processes, such as the transport of moisture from the soil to the atmosphere, and increases the work associated with energy costs for the formation of biological production.

## 4. Discussion

Let us compare our results with the results of other boreal forests, namely with the results of a similar analysis of the boreal ecosystems of the European plain (56.30 N, 32.53 E Central Forest Reserve, CFR) [52] and the boreal forests of the north-eastern Baikal region (55.35 N, 109.81 E Baikal region, BR) [72]. These ecosystems form a gradient along the absolute height and degree of continentality. The CFR has an altitude of 250 m above sea level, and has a temperate continental climate, flat topography, and, therefore, excess moisture most of the year, while the BR is located in an area of sharp continental climate, moderated by the influence of Lake Baikal in the summer, with elevations of 480–1240 m above sea level, compared to 1600–3000 m above sea level in Horidol Saridag (HS). The areas of the analyzed regions are comparable. It is also worth noting that the calculations for BR were carried out within the framework of the BGS model, without estimating the q-index, while calculations for CFR were carried out in two systems [53,73], and it was shown that the relationship between spatiotemporal variation in energy balance variables is generally similar between NSM and BGC. However, the variance in the variables within the NSM is greater, indicating a potentially greater sensitivity to diversity. Puzachenko and co-authors proposed converting the differences between the variables in the BGS and NSM systems to additional variables characterizing the “efficiency of self-organization” [52,53] and using extensive field material shows the promise of this approach. Our case located, in a territory different from the aforementioned, demonstrates that heat consumption for transpiration (exergy) generally has a seasonal course, strictly determined by the incoming solar radiation for all ecosystems, and that this pattern is universal; however, there are significant differences for different types of ecosystems. It is noteworthy that during the period of maximum vegetation activity for HS, meadow steppes, on average, have a level of evapotranspiration almost at the level of forests. Analysis according to the proposed scheme has always demonstrated a significant excess of exergy for forested areas over the exergy of treeless areas, including the monsoon deciduous forests of South Vietnam [74]. At this stage of research, we can only assume that this unusual situation is explained by a decrease in exergy in forests due to the summer minimum of precipitation. The behaviour of meadow systems as a whole is very typical and coincides in all regions. The differences in the dynamics of self-organization and the increment of information between larch and pine forests HS have no analogs for other territories. It is noteworthy that, in summer, larch forests have greater non-equilibrium and productivity (Figure 7c,d) and pine forests have greater self-organization (Figure 7f). This clearly indicates different mechanisms of self-regulation. Since pine forests are confined to the southern slopes, and accordingly experience a greater lack of moisture in the summer, the mechanism shown for CFR of adaptation of ecosystems to hot weather through increased self-organization and reduced evaporation costs in the pair “pine forests–larch forests” does not work. Apparently, there are other mechanisms at work here that are not yet known to us. Notably, only relationships within structural informational parameters (entropy, q-index, and information increment) diverge. In the southern taiga ecosystems, an observed pattern emerges: when the information increment increases, the entropy of reflected solar radiation decreases. Additionally, a robust positive linear correlation between informational increment and NDVI is evident: higher values of the former correspond to higher values of the latter. However, such connections do not manifest in mountain taiga ecosystems. Comparing the values of summer order parameters reveals disparities in their performance.

The interconnections between thermodynamic variables and vegetation properties within the Khövsgöl study area mirror patterns observed in other research areas. For instance, an increase in woody vegetation mass amplifies evapotranspiration costs. Interestingly, this increase has minimal impact on active surface temperature. Notably, the increments of information and productivity in flat landscapes tend to decrease with rising tree biomass. However, the reverse processes are observed in the Khövsgöl study area. A negative linear correlation between q-index and woody vegetation mass is noted: as q-index rises, woody vegetation mass diminishes in the Khövsgöl area, aligning with higher q-index values in meadows and lower values in the climax spruce forests of the European plain [53]. Considering the factor structure analysis (allocation of summer order parameters based on the invariants of all variables), it becomes apparent that q-index, the increment of information, and productivity index are determined by a singular order parameter. This parameter is linked to the variables on which the order parameter was calculated. Thus, higher NDVI and information increment values correspond to larger order parameters. Importantly, this relationship inversely impacts the q-index. Therefore, it can be asserted that the q-index, the increment of information, and the productivity index describe distinct mechanisms of vegetation feedback on changing weather conditions. Indeed, nearly all variables in the region exhibit high sensitivity to weather fluctuations, especially precipitation (humidity). Given the significant influence of q-index and information increment in classifying thermodynamic ecosystem types, these parameters potentially hold practical importance in identifying ecosystems with varying self-regulation scales, thereby contributing to landscape sustainability assessments.

Consequently, based on the results of the spatial–temporal analysis of thermodynamic variable changes in the Khövsgöl study area, several conclusions can be drawn. Firstly, the thermodynamic system operates in three stable states: summer, winter, and transitional. Secondly, three primary processes govern the system: “evapotranspiration”, “non-stationarity–productivity”, and “heat dissipation”. Thirdly, the coefficient of determination (R²) between relief and thermodynamic parameters is approximately 30–40%, with the most significant effect observed for structures around 570 m in linear size. Finally, system productivity increases with the rising Leaf Area Index (LAI) in the summer season and with the growth of stand basal area during the transitional period.

Two primary processes, “heat dissipation–evapotranspiration” and “non-stationarity–productivity”, were identified in ecosystems from CFR and the Baikal region. In contrast, three sets of processes, namely “evapotranspiration”, “non-stationarity–productivity”, and “heat dissipation”, were discerned in the Khövsgöl region. In summary, a hypothesis emerges: the temperature regime in the Khövsgöl region is notably less influenced by incoming solar radiation than other biomes, being primarily shaped by prevailing air masses. To test this hypothesis, relief contributions to the thermodynamic invariant of the summer heat field for both the Baikal and Khövsgöl regions were analyzed. The results revealed a coefficient of determination (R²) of 0.63 in both cases. In 2022, five temperature loggers were strategically placed along a 1 km transect to assess the impact of thermal regimes on solar radiation conversion. We anticipate obtaining results in 2024. It is evident that air humidity and precipitation play pivotal roles in the solar energy conversion process. Unfortunately, securing financial support for instrumental assessments of their long-term dynamics remains an ongoing challenge.

## 5. Conclusions

The research presents an intriguing prospect as it delves into the analysis of thermodynamic characteristics within the highly continental climate of the Khövsgöl region. Surprisingly, the results diverge from our previous findings, raising questions about the singular functioning of the Khövsgöl mountain ecosystem. This discrepancy leads to the formulation of a hypothesis, suggesting that these distinctive features may be attributed to a combination of factors, including weather conditions, orography, and climate. The research holds promise due to its potential to assess long-term variable dynamics in the future. The remarkably continental climate has enabled the accumulation of abundant cloudless Landsat scenes 4, 5, and 7 spanning from 1986 to 2011 (62 scenes). Our upcoming endeavors involve establishing correlations between thermodynamic variables and weather conditions using data obtained from the nearest weather station. Temperature data collected from loggers placed across the slope (in 2022) will be instrumental in validating and comparing meteorological data from the station situated in the basin. It is noteworthy that remote sensing temperature measurements often align well with data from weather stations, especially in flat southern taiga landscapes [68].

The applied information–thermodynamic approach to interpreting remote sensing data enables the quantification of thermodynamic variables within ecosystems. These calculated parameters hold practical significance in identifying ecosystems with varying degrees of self-regulation. For instance, the exergy of solar radiation provides an estimation of heat costs for evaporation, closely related to phytomass growth. Simultaneously, the q-index unveils the ecosystem’s capacity for self-organization. The information increment, often associated with NDVI, serves to detect the system’s non-stationarity and to assess potential ecosystem productivity. Ultimately, specific combinations of thermodynamic variables are intricately linked to ecosystem structure, governed by factors, such as vegetation cover, soil properties, and the hydrothermal regime. These thermodynamic variables prove invaluable in bridging the gap between field-obtained descriptions and instrumental measurements of ecosystem properties at specific points along the slope and across the entire area. This innovative approach facilitates the creation of maps with known error margins and limitations.

Employing this approach to assess complex, self-organizing ecosystems in diverse landscape and climatic conditions will enable a comprehensive understanding of their functioning within a unified measurement system. The ecosystems of Mongolia serve as representative models for numerous regions in Central Asia. Therefore, this study stands as a valuable contribution from both methodological and practical perspectives, offering insights into the intricate dynamics of ecosystems in this region. Since the warming rates in Mongolia exceed the global average, the issue of the stability of forest ecosystems is particularly pressing. Austrian researchers demonstrated through modelling [75] that topographic complexity enhances the stability of boreal forest ecosystems during warming. We believe that our research assessing the impact of various factors on the functioning of mountain landscapes will further enhance our understanding of the mechanisms of their regulation and, consequently, their stability.

## Figures and Tables

**Figure 1 entropy-25-01653-f001:**
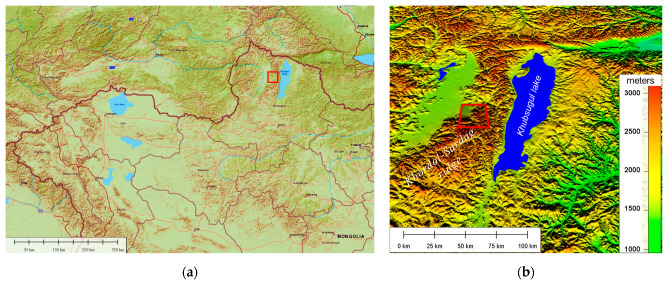
Study area (red square): (**a**) geographical position (World Street Map). (**b**) Digital elevation model SRTM (30 × 30 resolution) of the study area region.

**Figure 2 entropy-25-01653-f002:**
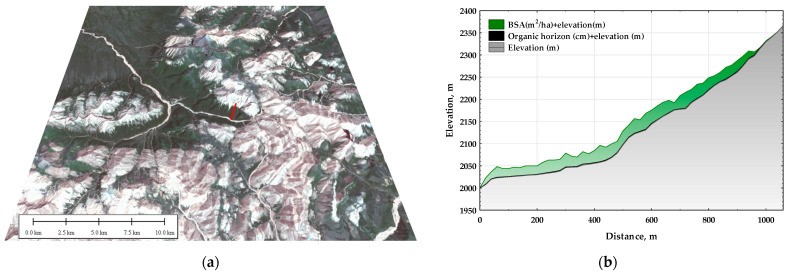
Sample plot area: (**a**) Landsat 8 OLI TIRS image 20 July 2020 on the digital elevation model (red line—transect). (**b**) Transect with 20 m sampling step (BSA—basal stand area of all live trees in a stand, meters per hectare, range for transect 0–37 m).

**Figure 3 entropy-25-01653-f003:**
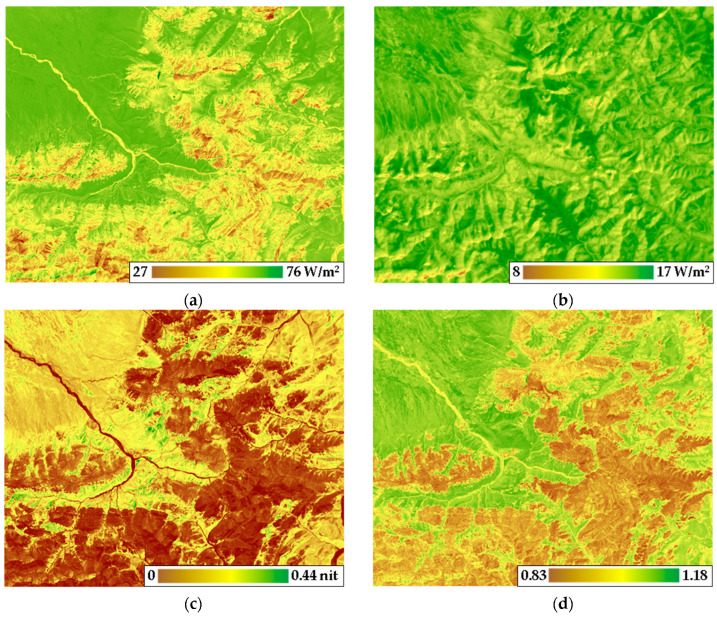
Summer order parameters for thermodynamic variables: exergy of solar radiation (**a**), bound energy (**b**), Kullback information increment (**c**), and q-index (**d**).

**Figure 4 entropy-25-01653-f004:**
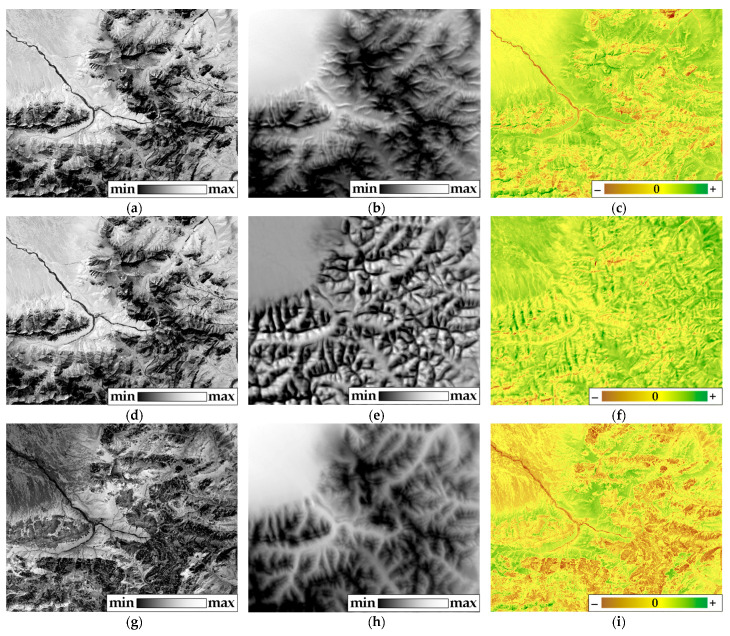
Order parameters of thermodynamic system (nondimensional) and the predicted meanings by morphometrical features: absorbed energy and exergy of solar radiation (**a**), predicted and (**b**) residuals (**c**); bound energy and temperature (**d**), predicted (**e**) and residuals (**f**); bioproductivity and structure (**g**), predicted (**h**) and residuals (**i**).

**Figure 5 entropy-25-01653-f005:**
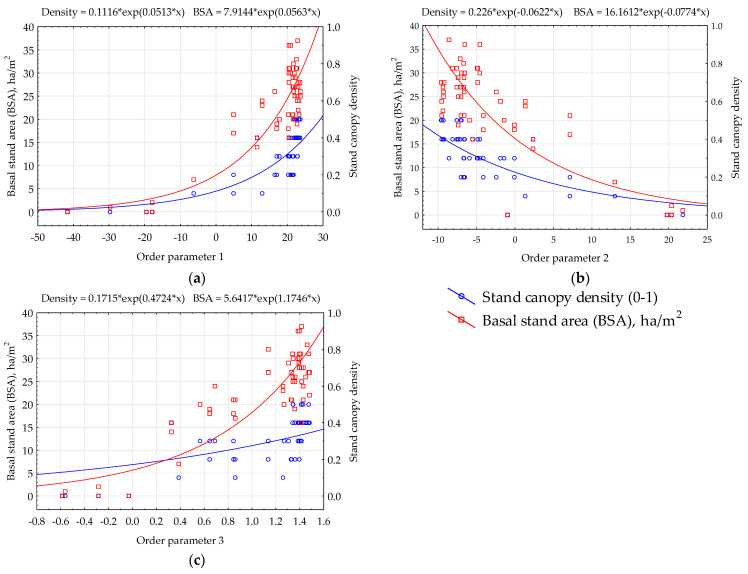
Order parameters of thermodynamic system (nondimensional) and vegetation cover properties: (**a**) order parameter 1—absorbed solar radiation and exergy; (**b**) order parameter 2—energy dissipation (heat flux and entropy); (**c**) order parameter 3—biological production.

**Figure 6 entropy-25-01653-f006:**
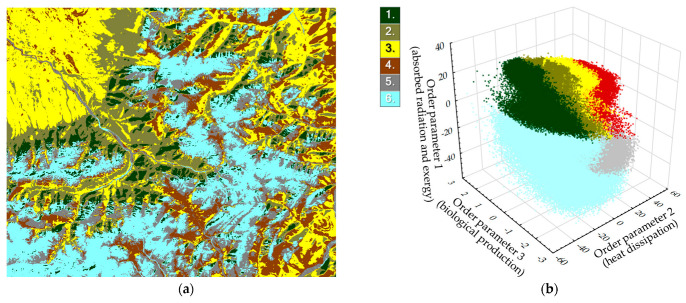
Types of thermodynamic systems: 1. larch forests; 2. pine forests; 3. meadow steppe; 4. grassland meadows; 5. open alpine grassland; 6. screes (based on [59]. (**a**) Classification. (**b**) Types in space of order parameters (dimensionless).

**Figure 7 entropy-25-01653-f007:**
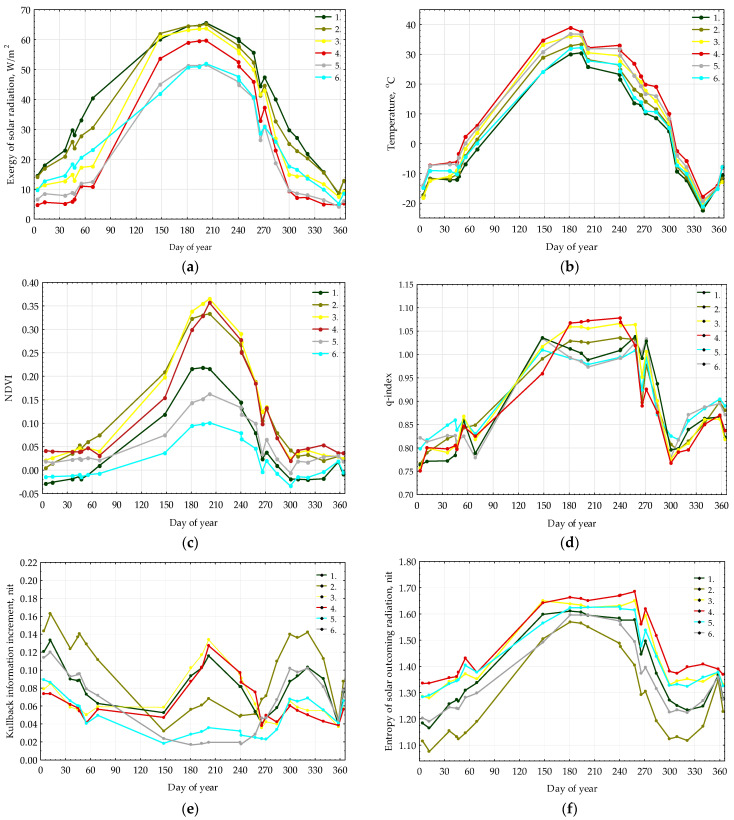
Seasonal dynamic of main thermodynamic variables for types of thermodynamic systems: (**a**) exergy; (**b**) temperature; (**c**) NDVI; (**d**) q-index; (**e**) Kullback information increment; (**f**) entropy of solar outcoming radiation. Type numbers detailed in Figure 6 description.

**Table 1 entropy-25-01653-t001:** Parameters of the used Landsat 8 OLI TIRS data.

Month	Day	Year	Day of Year (DOY)	Albedo *	Temperature, °C *
January	4	2018	4	0.46	−16.1
	12	2015	12	0.38	−10.1
February	5	2018	36	0.43	−9.6
	14	2018	45	0.42	−9.1
	16	2016	47	0.43	−7.5
	24	2019	55	0.34	−2.3
March	10	2021	69	0.39	2.6
May	28	2018	148	0.16	29.6
June	30	2015	181	0.13	34.6
July	13	2014	194	0.12	34.6
	20	2020	202	0.11	29.6
August	28	2019	240	0.11	28.7
	29	2017	241	0.11	26.7
September	15	2014	258	0.12	20.2
	22	2020	266	0.20	17.9
	27	2016	271	0.15	15.2
October	11	2015	284	0.25	13.5
	27	2021	300	0.40	6.9
November	5	2015	309	0.43	−6.0
	17	2017	321	0.40	−9.3
December	6	2018	340	0.46	−16.1
	24	2015	358	0.38	−10.1
	30	2018	364	0.43	−9.6

* Measured on Landsat bands.

**Table 2 entropy-25-01653-t002:** Order parameters for thermodynamic variables (23 terms) obtained by PCA. Explained variance ratio.

Thermodynamical Variables	Parameter 1	Parameter 2	Parameter 3	Total
	Winter (November–March)	Summer (June–September)	Mid-Season (May, October)	
Albedo	67.39	13.78	5.62	86.79
Absorbed radiation	66.84	14.96	5.88	87.05
Exergy	65.29	16.71	5.37	87.37
Bound energy	70.65	13.73	5.59	89.97
Internal energy increment	54.07	25.52	4.80	84.39
Temperature (heat flux)	67.10	17.66	-	84.76
NDVI	69.87	16.59	5.42	91.88
Kullback information increment	50.43	25.35	7.62	83.40
Entropy of solar outcoming radiation	64.28	15.65	5.69	85.62
q-index	40.65	22.10	9.00	71.75

**Table 3 entropy-25-01653-t003:** Contribution of morphometric characteristics of relief to the description of summer order parameter of thermodynamic variables. (R^2^—coefficient of determination of multiple regression, in cells—sign of correlation and linear dimensions of hierarchical levels. Red—positive relation, blue—negative relation.

Order Parameters of Thermodynamic Variables (Vegetation Period)	R^2^	Relief Morphometrically Parameters					
		Absolute Elevation	Slope	Curvature		Shaded Relief	
				Minimal	Maximal	South	East
Albedo	0.33	+ ^1^	−1890, 570	−570		1110	
Absorbed radiation	0.31	–		570			
Exergy	0.48	–			570		
Bound energy	0.33				−570, −1380	570	1380
Internal energy increment	0.51	+	−1890				150
Temperature (heat flux)	0.63	–	1110			1110	
NDVI	0.37	–			−1110	570	
Kullback information increment	0.30	–	−1890, 570			150	
Entropy of solar outcoming radiation	0.09				−570	570	1890
q-index	0.31	+	150		570	−570	

^1^ Sign of correlation.

**Table 4 entropy-25-01653-t004:** Correlations (r) between summer order parameters for each variable and the main ecosystem properties measured on a transect (20 m sampling step, 54 points, Figure 2). Red—positive relation, blue—negative relation.

Order Parameters of Thermodynamic Variables (Vegetation Period)	Ecosystem Properties						
	Basal Stand Area (BSA)	Leaf Area Index (LAI)	Stand Canopy Density	Grass-Shrub Layer Cover	Dry Biomass	pH	Organic Horizon Thickness
Albedo	− 0.81	−0.49	− 0.72	−0.13	−0.44	−0.48	−0.30
Absorbed radiation	0.83	0.50	0.72	0.11	0.42	0.49	0.30
Exergy	0.85	0.51	0.73	0.08	0.39	0.50	0.31
Bound energy	− 0.85	−0.48	− 0.67	0.32	0.05	− 0.56	−0.18
Internal energy increment	− 0.85	−0.47	− 0.76	−0.15	−0.37	− 0.51	−0.33
Temperature (heat flux)	0.07	0.26	−0.28	−0.59	−0.32	0.25	−0.28
NDVI	0.18	0.50	0.08	0.20	0.22	0.41	0.03
Kullback information increment	0.81	0.63	0.53	−0.20	0.06	0.59	0.16
Entropy of solar outcoming radiation	− 0.79	−0.29	− 0.78	0.07	−0.09	− 0.51	−0.28
q-index	− 0.85	− 0.61	− 0.64	0.11	−0.11	− 0.58	−0.21

**Table 5 entropy-25-01653-t005:** Correlations (r) between the summer order parameters of the landcover thermodynamic system and main ecosystem properties measured on a transect (20 m sampling step, 54 points, Figure 2). Value—order parameter meanings, residuals—regression residuals from relief. Red—positive relation, blue—negative relation.

Ecosystem Properties	Order Parameters of the Thermodynamic System					
	1		2		3	
	Value	Residuals	Value	Residuals	Value	Residuals
Basal stand area (BSA)	0.86	0.89	− 0.82	− 0.87	0.89	0.84
Leaf area index (LAI)	0.47	0.52	−0.33	−0.42	0.54	0.58
Stand canopy density	0.77	0.78	− 0.78	− 0.74	0.72	0.54
Grass-shrub layer cover	−0.16	−0.12	0.27	0.20	−0.15	−0.09
Dry biomass	0.38	0.32	−0.28	−0.2	0.17	−0.01
pH	−0.32	−0.26	0.29	0.15	−0.11	0.1
Organic horizon thickness	0.33	0.31	−0.30	−0.23	0.24	0.13

## Data Availability

Original Landsat multispectral imagery scenes are available through Google Earth Engine (GEE). Macros (script) for GEE for selecting and calculating variables are available upon request. Field data available upon request.

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
