# Peer review of "Multispectral Remote Sensing Data Application in Modelling Non-Extensive Tsallis Thermodynamics for Mountain Forests in Northern Mongolia"

_entropy, 2023, doi:10.3390/e25121653_

Round 1

Reviewer 1 Report

Comments and Suggestions for Authors

-Ecosystem functioning and landscape dynamics details with respect to thermodynamcs approach can be explained more detailed for this mountain,

​It should be explained how appropriate your thermodynamic classification in Figure 6 is in terms of forestry science; 

- Nomenclature must be added; i.e. at Eq.8. and Eq.9 e is spesific exerg or ln base; must be explained;

- Eq.7 and Eq.8 must be explained more detailed; independent and dependent variables meanings musr be clarified

- Fig.8 must be more clear and understandable;

Author Response

Reviewer 1

Dear Reviewer,

thank you for your favourable feedback. Your questions and comments have been highly valued. We did our best to address all the questions you raised. Please, find below our point-by-point answers.

-Ecosystem functioning and landscape dynamics details with respect to thermodynamcs approach can be explained more detailed for this mountain,

Thank you for the recommendation. The text contextualizing the study and highlighting the ecosystem functioning and landscape dynamics details has been inserted into the Introduction at lines 160-185 Also we have described the “Results” and “Discussion” sections in detail.

- It should be explained how appropriate your thermodynamic classification in Figure 6 is in terms of forestry science;

Thank you for the suggestion.

Our types are presented in accordance with the "Vegetation map of Mongolian People's Republic" [CrossRef] and the "Ecosystems of Mongolia. 2019. Atlas." [https://www.researchgate.net/publication/337830628_ECOSYSTEMS_OF_MONGOLIA_ATLAS]. Vegetation types were interpreted based on our field geobotanical descriptions and visual interpretation of satellite images. Since we do not extensively discuss the correlation of thermodynamic characteristics with parameters of plant communities (e.g., such as species cover) and soils, except for BSA and height, we did not specify the dominants of the understory in this case. However, they are detailed in Section 2.1. Study Area (lines 213-218). We give this comment at lines 553-567.  

- Nomenclature must be added; i.e. at Eq.8. and Eq.9 e is spesific exerg or In base; must be explained;

Thank you for the suggestion, the clarifications have been made at line 318 (about eq 7 and 8) since equation 9 relies on 7 and 8, we did not explain it separately.

- Eq.7 and Eq.8 must be explained more detailed; independent and dependent variables meanings must be clarified

Thank you for the suggestion, the clarifications have been made at lines 310-314

- Fig. 8 must be more clear and understandable

Thank you for the suggestion. The article contain 7 figures. Perhaps Reviewer means figure 6.

Unfortunately, we don’t understand what exactly we could improve about it. In our experience, the problem every time (our previous work in entropy with similar figures (links 73,74) is always caused by the dimension of the order parameters (the axes of figure 6, (y,x,z). Since they generalize various variables, they actually are dimensionless. We have made an appropriate explanation for the figure.

Reviewer 2 Report

Comments and Suggestions for Authors

The paper "Thermodynamic analysis of Northern Mongolia mountain forest ecosystems (based on remote sensing data)" presents analysis of forest ecosystems using remote sensing data to assess the fluctuations in energy and organizational parameters within the mountain taiga-meadow landscape by applying Tsallis statistics concept. The title should include a term such as "Tsallis statistics" or "non-extensive statistics" to be more appropriate. Furthermore, the parentheses in the title appear to be inappropriate. Although the paper deals with an exciting subject, the presentation of this work needs to be improved since there are missing or poorly motivated definitions. The equations must also be better written and described.  Please see the comments below. 

1) What is Tsallis non-extensive thermodynamics and how is it applied in this study? This question must be answered in the introduction of the manuscript.

2) The statements made in lines 96-109 need to be supported by adequate references.

3) Please consider changing "Figure 1a,b" to "Figure 1" in line 172.

4) Please include a brief explanation of the energy balance calculation used, the one proposed by Yury Puzachenko et al.

5) The mathematical notation must be completely revised. For example, in line 247, "implies energy reflected in spectral band v to the total energy", it is recommended to write the variable v with the same font as the equations. Still in this line, capital P should actually be lowercase p. Besides, “e” represents an exponential function of “in”? Is not it is an exponential function? Please clarify these points.

6) Equations (7) and (8) are only valid when the condition of y(x=0) = 1. The authors must include such information.

7) In line 284, the authors use the variable p for both probability and a free parameter. Also in this line, notice that there is a typing error, p>0p>0.

8) What is the range of variability q?

9) In lines 285-286, the authors state that "A higher q signifies greater internal correlations and a more organized system. ". References and examples are necessary to support such a statement. Indeed, the entire remaining paragraph must be adequately referenced.

10) Equation (12) represents the Kullback–Leibler divergence in the Tsallis statistics. A wide range of works have already proposed it, such as in https://doi.org/10.1016/j.jmaa.2015.12.008. Since such a definition is not proposed in this work, appropriate references must be introduced.

Comments on the Quality of English Language

A proofreading is necessary for this manuscript. Several grammatical and typing errors are throughout the text, including issues with the title of section 2.2. Please carefully review the entire manuscript for such errors and consider a thorough proofreading.

Author Response

Dear Reviewer,

We would like to express our gratitude for your positive response to our manuscript. We appreciate your recognition of the importance of our work, as well as the valuable comments you have provided. The title, definitions and equations have undergone careful revision. Also, we corrected the linguistic imperfections thorough the entire manuscript. Also we have described the “Results” and “Discussion” sections in detail.  We apologize for any inconvenience caused by the layout of the article. We have made our best efforts to address all the questions you raised, and we present our point-by-point answers below. Since we made a large amount of changes, we have not yet resorted to editorial proofreading. If you decide that our text does not require any more serious revision, we will contact MDPI proofreading.

1) What is Tsallis non-extensive thermodynamics and how is it applied in this study? This question must be answered in the introduction of the manuscript.

Thank you for the suggestion, the information has been provided in the Introduction in lines 120 – 159 lines
2) The statements made in lines 96-109 need to be supported by adequate references.

Thank you for the suggestion, the references have been provided. In 98-102 lines. We would like to note that in this case we were not referring to the results of research, but to the fundamentals of reflection of solar radiation that underlie remote sensing

3) Please consider changing "Figure 1a,b" to "Figure 1" in line 172.

Thank you for the suggestion, the change has been made – line 190

4) Please include a brief explanation of the energy balance calculation used, the one proposed by Yury Puzachenko et al.

Thank you for the recommendation. The required text has been incorporated into the Materials and Methods section. Lines 247 - 269

5) The mathematical notation must be completely revised. For example, in line 247, "implies energy reflected in spectral band v to the total energy", it is recommended to write the variable v with the same font as the v in this line, capital P should actually be lowercase p. Besides, “e” represents an exponential "in"these points.

Thank you for the recommendation. All the mathematical notations have been revised throughout the entire manuscript.

1) We decided to clarify the notations in equation (2) in more detail. 2) We have given the notation of this equation in accordance with our work in Entropy [https://doi.org/10.3390/e22101132]. 3) We believe there was some confusion with the upper indices denoting the input and output radiation. We hope the explanations made in lines 277 - 280 will help to clarify the issue.

6) Equations (7) and (8) are only valid when the condition of y(x=0) = 1. The authors must include such information.

Thank you for the comment. Special notes concerning this condition have been added at line 318.

7) In line 284, the authors use the variable p for both probability and a free parameter. Also in this line, notice that there is a typing error, p>Op>0.

Thank you for the comment. The sections have been revised (line 324)

8) What is the range of variability q?

The q-index can take any value. In our measurements, it is usually from 0 to 2

9) In lines 285-286, the authors state that "A higher q signifies greater internal correlations and a more organized system.". References and examples are necessary to support such a statement. Indeed, the entire remaining paragraph must be adequately referenced.

Thank you for the suggestion, the references have been provided in line 332

10) Equation (12) represents the Kullback-Leibler divergence in the Tsallis statistics. A wide range of works have already proposed it, such https://doi.org/10.1016/j.jmaa.2015.12.008. Since such a as in definition is not proposed in this work, appropriate references must be introduced.

Thank you for the suggestion, the references have been provided. Line 339

Round 2

Reviewer 2 Report

Comments and Suggestions for Authors

The authors addressed all points raised in the first round of review.